# Is the Triggering of PD-L1 Dimerization a Potential Mechanism for Food-Derived Small Molecules in Cancer Immunotherapy? A Study by Molecular Dynamics

**DOI:** 10.3390/ijms24021413

**Published:** 2023-01-11

**Authors:** Xiaoyan Wu, Na Wang, Jianhuai Liang, Bingfeng Wang, Yulong Jin, Boping Liu, Yang Yang

**Affiliations:** Key Laboratory for Bio-Based Materials and Energy of Ministry of Education, College of Materials and Energy, South China Agricultural University, Guangzhou 510630, China

**Keywords:** PD-1/PD-L1 pathway, food-derived molecules, inhibitor drugs, molecular docking, molecular dynamics simulation

## Abstract

Using small molecules to inhibit the PD-1/PD-L1 pathway is an important approach in cancer immunotherapy. Natural compounds such as capsaicin, zucapsaicin, 6-gingerol and curcumin have been proposed to have anticancer immunologic functions by downregulating the PD-L1 expression. PD-L1 dimerization promoted by small molecules was recently reported to be a potential mechanism to inhibit the PD-1/PD-L1 pathway. To clarify the molecular mechanism of such compounds on PD-L1 dimerization, molecular docking and molecular dynamics simulations were performed. The results evidenced that these compounds could inhibit PD-1/PD-L1 interactions by directly targeting PD-L1 dimerization. Binding free energy calculations showed that capsaicin, zucapsaicin, 6-gingerol and curcumin have strong binding ability with the PD-L1 dimer, where the affinities of them follow the trend of zucapsaicin > capsaicin > 6-gingerol ≈ curcumin. Analysis by residue energy decomposition, contact numbers and nonbonded interactions revealed that these compounds have a tight interaction with the C-sheet, F-sheet and G-sheet fragments of the PD-L1 dimer, which were also involved in the interactions with PD-1. Moreover, non-polar interactions between these compounds and the key residues Ile54, Tyr56, Met115 and Ala121 play a key role in stabilizing the protein–ligand complexes in solution, in which the 4′-hydroxy-3′-methoxyphenyl group and the carbonyl group of zucapsaicin, capsaicin, 6-ginger and curcumin were significant for the complexation of small molecules with the PD-L1 dimer. The conformational variations of these complexes were further analyzed by free energy landscape (FEL) and principal component analysis (PCA) and showed that these small molecules could make the structure of dimers more stable. This work provides a mechanism insight for food-derived small molecules blocking the PD-1/PD-L1 pathway via directly targeting the PD-L1 dimerization and offers theoretical guidance to discover more effective small molecular drugs in cancer immunotherapy.

## 1. Introduction

Programmed cell death-1 (PD-1) [1,2,3], as an immune checkpoint protein expressing on the surface of activated T cells, plays a fatal role in cancer immunotherapy when interacted with programmed cell death ligand-1 (PD-L1) [4]. Cancer cells release PD-L1, which binds to PD-1 and leads to the immune escape of cancer cells [5]. Blocking the immune checkpoint with inhibitor drugs is the most encouraging strategy with significant advantages over conventional chemotherapy [6]. Compared with the monoclonal antibodies (mAbs) that have been widely applied in clinical trials, the small molecular drugs are starting to attract attention by the characteristic of higher stability, better tumor penetration and fewer side effects [7]. The pioneering work was made by the Bristol Myers Squibb (BMS) company, who synthesized a series of small molecules (BMS-8, BMS-200, BMS-202, etc.) and found out that they inhibit the PD-1/PD-L1 pathway by inducing the dimerization of PD-L1 [8]. In particular, the crystal structure of PD-1/PD-L1 obtained by Zak et al. in 2015 offered druggable pharmacophore design models for developing more reliable and accessible small molecules targeting PD-L1 [9].

Molecular simulation has become a powerful and reliable technique of characterizing the detailed molecular interaction mechanisms, which could greatly accelerate drug development [10]. This technique has also been applied in studying the interaction mechanism between BMS molecules and the PD-L1 dimer by several researchers [11,12,13,14,15,16,17,18]. These works revealed that BMS small molecules could bind tightly with the hydrophobic pocket of the PD-L1 dimer, and Ile-54, Tyr-56, Met-115, Ala-121, and Tyr-123 were confirmed to be the key residues in binding modes. Our research group studied the effects of the chirality of BMS molecules on inhibitory activities via conducting molecular dynamics simulation between the modified and unmodified BMS molecules during the ligand recognition process. Meanwhile, the modified BMS molecules bound to key residues of the PD-L1 dimer more tightly than unmodified molecules, and the former generated more H-bonds than the latter [12]. Furthermore, some approved drugs based on the biphenyl skeleton similar to the structure of BMS molecules, such as Flurbiprofen analogues, Pyrvinium and Pyrazolones, were proved to have the ability of targeting the PD-L1 dimer and blocking the PD-1/PD-L1 pathway [19,20,21]. The molecular dynamics simulation and binding energy calculation demonstrated that the studied approved drugs with biphenyl skeleton showed a similar binding mode with the PD-L1 dimer to those of BMS molecules. Using molecular simulation to screen efficient small molecule drugs targeting the PD-1/PD-L1 pathway is promising but still faces a lot of unknowns and needs to be further explored. One of the most urgent questions to clarify is whether other anticancer small molecules beyond the BMS structure downregulate the expression of PD-L1 by acting on PD-L1 dimers.

In addition to the man-made small molecules, natural compounds have received more and more attention in the development of small molecular drugs due to low toxicity, good biocompatibility and strong penetrability [22,23]. Particularly, many food-derived molecules have been reported to downregulate the expression of PD-L1 in cancer immunity [23]. Capsaicin, an alkaloid found in the capsicum, is widely considered to have a great effect in antitumor therapy and cancer immunotherapy [23,24]. Maria et al. [25] found that capsaicin can downregulate PD-L1 expression in cancer cells in renal cell carcinoma. Zucapsaicin, a cis-isomers of capsaicin, is reported to be a potential anticancer molecule [26]. 6-Gingerol is a natural phenolic compound derived from the root of ginger which has a remarkable effect on a variety of cancers [27]. Sp et al. [28] found that 6-gingerol could act on embryonic cancer cells and inhibit its activity via downregulating the expression of PD-L1. Although the above work suggests that many food-derived small molecules showed significant anticancer effects, the mechanism of these molecules interacting with PD-L1 remains unclear.

For food-derived small molecules, only a few molecular simulation studies focused on the interaction mechanism between the small molecules and the PD-L1 dimer have been reported. Verdura et al. experimentally found that resveratrol could downregulate PD-L1 expression and demonstrated that the induction of PD-L1 dimerization was involved via molecular simulation [29]. More recently, our group presented a molecular-level picture of the inhibitory mechanism of food-derived polyphenols, including curcumin, resveratrol and epigallocatechin gallate, by using a wide array of computational approaches [30]. It was found that the curcumin, resveratrol and epigallocatechin gallate have a similar binding mode with the BMS series, namely to block the PD-1/PD-L1 pathway by inducing PD-L1 dimerization. The curcumin system showed the lowest binding free energy among the three polyphenols. As the feature structure of curcumin, the carbonyl and 4′-hydroxy-3′-methoxyphenyl at the middle and ends of the long chain skeleton were considered to make a significant contribution in the combination of curcumin with the PD-L1 dimer [30]. We suspect that the other small molecules embedded with the same feature groups as curcumin, such as capsaicin, zucapsaicin, and 6-gingerol (Figure 1a,b,d) which have anti-cancer properties, would bind to the PD-L1 dimer by occupying the channel-like hydrophobic pocket. Meanwhile, by searching for small molecules with lower binding free energy with the PD-L1 dimer, more effective small molecules drugs might be developed.

Based on the above viewpoint, the present study aimed to screen for more effective small molecule inhibitors potentially applied for cancer immunotherapy by using molecular simulation techniques. Three natural compounds (capsaicin, 6-gingerol, and curcumin as a reference) and one approved drug (zucapsaicin), bearing the same feature groups of the carbonyl and 4′-hydroxy-3′-methoxyphenyl, were selected (Figure 1). In the current investigation, the molecular mechanisms of capsaicin, zucapsaicin, curcumin, and 6-gingerol targeting on the PD-L1 dimer were performed by molecular docking, molecular dynamics simulation and binding free energy calculations. Firstly, the three-dimensional structure of capsaicin (ZINC1530575), curcumin (PDB ID: 4K58), zucapsaicin (ZINC4468952), 6-gingerol (ZINC15331846), and the PD-L1 dimer separated from the X-ray crystal structure of the complex with the inhibitor BMS-200 (PDB ID: 5N2F), was selected as the initial model of ligands and receptor, respectively. According to the reported binding site by Guzik et al. [8], these small compounds and the PD-L1 dimer would form into complexes by using molecular docking. In addition, the restrained electrostatic potential (RESP) of all small molecules were computed for subsequent dynamics simulations [31]. Furthermore, nanosecond molecular dynamics simulations were performed for each complex system to illustrate the mechanism of action for small molecules at the binding site. Then, the molecular mechanics–Poisson Boltzmann surface area (MM-PBSA) method [32,33] and Mindist module were applied to explore the binding affinity of small molecules with the PD-L1 dimer and the receptor–ligand binding mode. Finally, principal component analysis (PCA) [34,35] and free energy landscape (FEL) [36,37,38] were utilized to describe the impact of these compounds on the binding site and the overall atomic motion of the PD-L1 dimer. Generally, we compared the binding affinity, binding mode and contribution of per-residues for each complex system. The binding affinity of small molecular systems follows the trend: zucapsaicin > capsaicin > 6-gingerol ≈ curcumin, under the consideration of interaction entropy. The global motion of dimers under the action of different small molecules was illustrated, and the interaction of ligands and key residues of these complex systems has been confirmed, in which Ile-54, Tyr-56, Gln-66, Met-115, and Ala-121 were identified as the key residues. Therefore, this work provides a computational basis for the discovery of new food-derived molecules to block PD-1/PD-L1 interaction. The studied small molecules and its derivative may serve as potential candidates in cancer immunotherapy.

## 2. Results and Discussion

### 2.1. Docking of PD-L1 Dimer and Small Molecules

By using molecular docking [39], the four small compounds (capsaicin, zucapsaicin, curcumin and 6-gingerol) and the PD-L1 dimer formed into protein–ligand complexes (Appendix A). Based on the top-ranked docking model for each complex, the strongest binding affinity in the channel-like pocket in a given direction was calculated. The structural models with the strongest binding affinity for the four different small molecules are shown in Figure 2 [40]. In addition, to check whether the docking method is reasonable, the original ligand (BMS-200) was used to redock with the PD-L1 dimer (5N2F), requiring that the root mean square deviation (RMSD) between the docked pose and the original one is less than 2 Å [41]. As a result, both the crystal and the docked structures overlapped within the cavity formed between PD-L1 chain A ((A)PD-L1) and PD-L1 chain B ((B)PD-L1), and the overlap showed an RMSD of 1.62 Å.

From the docking results of the PD-L1 dimer with capsaicin, zucapsaicin, 6-gingerol and curcumin, respectively, it can be seen that the binding of the four compounds mainly occurs in the inner channel pocket of the dimer (Figure 2). The studied four food-derived molecules have the similar binding modes with the original ligand (BMS-200), suggesting that the docking protocol could be utilized to identify the binding conformations of the capsaicin, zucapsaicin, 6-gingerol and curcumin systems. Then, the docked complexes were applied as initial coordinates in the following MD simulations. The method of building the initial model is referred to the work of Guo et al. [30], while all the charges of small molecules are calculated as RESP charge, considering that RESP charges are more accurate than BCC in dynamic simulation [31].

### 2.2. RMSD and $R_{g}$

The 200 ns MD simulation initialized from the PD-L1 dimer–small molecules complex system obtained from Section 2.1 was carried out. At the very beginning of this study, we have elongated the simulation time of the curcumin/PD-L1 dimer system to 500 ns and performed the analysis on the complex simulation. As shown in Appendix A, the complex system reached equilibrium state at 10 ns and kept this state for the subsequent 490 ns simulation. Considering that the data produced by 200 ns simulation is enough for all systems to reach equilibrium and longer simulation time would consume too much computing resources, all the complex systems in this study were simulated by 200 ns and repeated three times. The trends of RMSD and gyrate over time are shown in Figure 3. The RMSD value is an important profile to estimate the equilibration procedure in the simulation trajectory and the stability of protein structure upon the binding of the ligand.

Because the residues of the loop area are far away from the binding site and do not interact with small molecules, the RMSD of the residues within 20 Å of the binding site is computed as well as for the average RMSD in equilibrium for each system. The results indicated that the capsaicin, zucapsaicin, and 6-gingerol complex systems arrive at equilibrium within 5 ns simulation, and the average RMSD values for capsaicin, zucapsaicin, and 6-gingerol system are respectively 2.71, 2.41, and 2.49 Å when the simulation reaches equilibrium. However, it took 100 ns for curcumin and the dimer system to arrive at steady state, where the RMSD values of the PD-L1 dimer and curcumin complex system were 3.13 and 2.61 Å, respectively. Obviously, the RMSD value of the dimer system was higher than that of all the complex systems, indicating that the existence of these small molecules could stabilize the backbone of the dimer. It might also result from the bigger sport space and lower steric hindrance of the atoms in the dimer system where small molecules are absent [42].

Concurrently, as a powerful tool to describe the compact degree of protein in solvent, the radius of gyration ($R_{g}$) of the dimer system and the four complex systems were computed. As shown in Figure 3b, the $R_{g}$ value of the dimer system is lower than that of the complex systems, suggesting that the dimer system has a more compact structure than the complex ones in solvent.

### 2.3. RMSF

Root mean square fluctuation (RMSF) was used to measure the level of atoms fluctuation, and the RMSF results of the MD simulations are shown in Figure 4. The distributions of RMSF in each system are generally similar, while the C-sheet (54–56), F-sheet (115–117) and G-sheet (121–123) of the dimer system have higher fluctuation than the complex systems. These sheets were also the key sites for the combination of PD-1 and PD-L1. Based on the results of $R_{g}$ and RMSF, the dimer in solvent was squeezed during 200 ns simulation and the residues in the binding pocket were more flexible for the dimer system. Furthermore, the interposition of small molecules increased the rigidity of the dimer, since the PD-L1 dimer will close the binding channel in the absence of small molecules (Appendix A).

### 2.4. Binding Free Energy Calculation

Based on the MD simulation trajectories, binding free energies of small molecules to the PD-L1 dimer were calculated by using the MM-PBSA method ($\Delta G_{bind}=G_{complex}-G_{receptor}-G_{ligand}$) to analyze the binding affinities of the PD-L1 dimer and these small compounds [43], where each system was repeated three times to calculate the $\Delta G_{bind}$. In our previous work [30], the entropy was not added to the calculation of binding free energy, because it would consume a large quantity of computing resources and require a long computing time. However, the contribution of entropy to free energy cannot be ignored, especially for the comparison of systems with approximate free energy values. Recently, some researchers have reported that the IE method showed good efficiency in calculating the entropy change of ligand and protein [44], so the contribution of entropy to the binding free energy for each complex system has been calculated in this work by using the IE method. Thus, the binding free energies calculated in this work are more accurate than those in our previous work. The binding free energy and the contributions of its components for each complex system are summarized in Table 1. The binding free energy values of capsaicin, zucapsaicin, 6-gingerol and curcumin to the PD-L1 dimer are −42.53 kcal/mol, −43.75 kcal/mol, −39.20 kcal/mol and −39.19 kcal/mol, respectively.

The negative value of $\Delta G_{bind}$ for the complex systems indicated that the combination of dimer and ligands was a spontaneous process and the configuration of the complex could stably exist in the solvent. Meanwhile, the binding free energy can effectively evaluate the combination ability of receptor and ligand, in which the level of binding was corresponding to the level of biological activity. According to the calculation results, the binding capacity of these small molecules showed the following trend: zucapsaicin > capsaicin > curcumin ≈ 6-gingerol. Capsaicin and zucapsaicin showed much stronger binding affinity than 6-gingerol and curcumin, suggesting that capsaicin and zucapsaicin may serve as more effective potential drugs for cancer immunotherapy. Intuitively, the van der Waals term, the electrostatic term and the non-polar term have beneficial effects for the combination of small molecules and dimers, in which the van der Waals term plays a main role in the binding, whereas the polar term is disadvantageous for binding. Usually, $\Delta E_{vdmwaals}$ and $\Delta E_{nonpolar}$ are closely correlated with the hydrophobic interactions which could be linked to the burial of hydrophobic groups [45]. The $\Delta E_{vdmwaals}$ + $\Delta E_{nonpolar}$ of capsaicin, zucapsaicin, curcumin and 6-gingerol were −61.15 kcal/mol, −59.75 kcal/mol, −60.09 kcal/mol and −56.77 kcal/mol, respectively. This shows beneficial contributions for binding free energies, indicating that the hydrophobic interaction drives these compounds binding to the PD-L1 dimer. Furthermore, this result shows that the common structure of the 4′-hydroxy-3′-methoxyphenyl backbone in these small molecules has interacted with the hydrophobic channel binding site via van der Waals interactions, and the small molecules were squeezed into the hydrophobic channel due to the entropic effect of the solvent.

In addition, the RMSD of the PD-L1 monomer/small molecule system showed that small molecules could interact with the PD-L1 monomers as shown in Appendix A. Meanwhile, the binding free energies of the PD-L1 monomer/capsaicin, PD-L1 monomer/zucapsaicin, PD-L1 monomer/curcumin and PD-L1 monomer/6-gingerol systems were calculated and summarized in Table 2. Comparing the binding free energy of small molecules to the dimer and monomer (Table 1 and Table 2), the $\Delta G_{bind}$ of the PD-L1 dimer (−39.19 ± 2.55~−43.75 ± 2.58 kcal/mol) is more negative than the $\Delta G_{bind}$ of the PD-L1 monomer (−14.56 ± 5.52~−26.28 ± 3.47 kcal/mol) when the food-derived small molecules are present. This result further verified the conclusion that the PD-L1 dimer could be stabilized by the studied food-derived molecules.

### 2.5. Per-Residue Energy Decomposition and Contact Numbers

In order to characterize the binding regions of capsaicin, zucapsaicin, curcumin and 6-gingerol on the PD-L1 dimer, the residue energy is generated by the MM-PBSA method in gmxMMPBSA software, and the contact number was calculated between small molecules and residues, in which the residues within 5 Å of the binding site were considered (Table 3, Table 4, Table 5 and Table 6, Figure 5). Herein, the residues with binding energy contributions less than −1 kcal/mol were considered to be key residues. Meanwhile, the residues of more than 10 contacts were considered to play a significant effect on intermolecular interactions [46]. As depicted in Figure 6, capsaicin interacted with ten key residues: (A)ILE-54, (A)Tyr-56, (A)Met-115, (A)Ala-121, (B)Ile-54, (B)Val-55, (B)Tyr-56, (B)Gln-66, (B)Met-115, and (B)Ala-121. As shown in Figure 7, zucapsaicin interacted with ten key residues: (A)Tyr-56, (A)Arg-113, (A)Met-115, (A)Ala-121, (B)Ile-54, (B)Val-55, (B)Tyr-56, (B)Gln-66, (B)Met-115, and (B)Ala-121. As shown in Figure 8, 6-gingerol interacted with eight key residues: (A)ILE-54, (A)Tyr-56, (A)Met-115, (A)Ser-117, (A)Ala-121, (B)Ile-54, (B)Tyr-56, and (B)Gln-66.

In addition, curcumin interacted with eight key residues: (A)Ile-54, (A)Tyr-56, (A)Met-115, (A)Tyr-123, (B)Ile-54, (B)Tyr-56, (B)Met-115, (B)Ala-121 in the PD-L1 dimer in Figure 9, which were similar to the calculation made by Guo et al. [30]. From Table 3, Table 4, Table 5 and Table 6, these molecules almost occupy the target space of BMS-200 that binds to the PD-L1 dimer, which is consistent with the results of molecular docking. The binding free energy of these key residues with capsaicin and zucapsaicin is lower than that of curcumin and 6-gingerol. Meanwhile, capsaicin, zucapsaicin and 6-gingerol, in contrast to curcumin, are more inclined to bind to key residues of chain B in the binding cavity (see Appendix A).

In short, the four food-derived molecules studied in this work showed strong binding ability to the beta sheet of PD-L1, where the interaction of the C, F, G sheet and small molecules has the lowest binding energy and the highest contact number. The C sheet, F sheet and G sheet are the secondary structure that forms the binding pocket and the significant part of PD-1/PD-L1 interaction. Furthermore, the C sheet, F sheet and G sheet take an important place in the combination of PD-L1 and capsaicin, zucapsaicin and 6-gingerol, respectively. These results suggested that it is possible to further stabilize the binding pocket through structural modification of the small molecules.

### 2.6. Binding Mode Analysis

In order to characterize the structure variation of small molecules and key residues from 0 to 200 ns, the binding modes of small molecules in the binding pocket are analyzed by the PLIP program and VMD, in which only H-bonds donors and acceptors with an occupancy of more than 10% occupancy are shown in Table 7. It can be seen that the capsaicin and zucapsaicin have generated H-bonds with (B)Gln-66 and (A)Gln-66, respectively, in which the H-bonds occupancy between the carbonyl group of Gln-66 sidechain and hydroxyl group of capsaicin and zucapsaicin reached 91.40% and 65.87% (Table 7). Meanwhile, (A)Ile-54, (A)Tyr-56, (A)Gln-66, (A)Met-115, (A)Ala-121, (A)Tyr-123, (B)Ile-54, (B)Tyr-56, (B)Met-115, (B)Ala-121 and (B)Tyr-123 are the residues that can form a hydrophobic interaction with capsaicin, as shown in Figure 6. Zucapsaicin interacts with (A)Tyr-56, (A)Gln-66, (A)Met-115, (A)Ala-121, (A)Tyr-123, (B)Ile-54, (B)Tyr-56, (B)Met-115 and (B)Ala-121 by hydrophobic interaction (Figure 7). 6-Gingerol interacts with (A)Ile-54, (A)Tyr-56, (A)Val-68, (A)Met-115, (A)Ala-121, (A)Tyr-123, (B)Ile-54, (B)Tyr-56, (B)Val-68, (B)Met-115, (B)Ala-121 and (B)Tyr-123 by hydrophobic interaction; additionally, (A)Gln-66 and (A)Ser-117 produced H-bonds with 6-gingerol (Figure 8). For the curcumin system, (A)Met-115, (A)Ala-121, (A)Tyr-123, (B)Ile-54, (B)Tyr-56, (B)Gln-66, (B)Met-115 and (B)Ala-121 play an important role in stabilizing dimer by hydrophobic interaction (Figure 9).

The intermolecular interactions of these systems are directly corresponding to the binding energy calculation and per-residue energy decomposition, in which the $\Delta E_{ele}$ term is mainly derived from the residue contribution energy of Gln-66, Asp-122 and Ser-117 in complex systems. In addition, we calculated the H-bonds occupancy of these systems, in which the occupancy > 50% was considered as indicative of strong H-bonds. Capsaicin, zucapsaicin and 6-gingerol could generate strong H-bonds with Gln-66, whereas strong H-bonds existed between curcumin and Tyr-123. In general, the electrostatic interaction dominated the interaction energy of the conventional H-bond [47]. The $\Delta E_{ele}$ values of capsaicin, zucapsaicin, curcumin and 6-gingerol were −4.76 kcal/mol, −3.06 kcal/mol, −4.14 kcal/mol and −7.06 kcal/mol, respectively. The H-bonds occupancy and average numbers were consistent with the results obtained from the $\Delta E_{ele}$ value.

Based on the results of binding modes, it could be concluded that the hydrophobic interaction is the main driving force for small molecules binding to the PD-L1 dimer. Meanwhile, small molecules could induce (A)PD-L1 and (B)PD-L1 to move toward each other, making them bind more tightly at the binding site, which is consistent with the results of RMSF and the radius of gyration.

### 2.7. Cross-Correlation Matrix

A correlation coefficient matrix helps us to understand protein structure and becomes a powerful tool for disclosing atomic motion information. In order to further understand the role of small molecules acting on the protein structure change, we construct the dynamic correlation coefficient matrix from the residue of $C_{\alpha }$ on the correlation coefficient matrix of the atoms, which can reflect the detailed atomic dynamic state of the PD-L1 dimer. The translation and rotation of atoms in trajectories were eliminated to prevent the influence of atomic motion [48].

The color depth of red areas and blue areas reflected the degree of $C_{\alpha }$ atoms’ anti-correlated and correlated motions, respectively, which represent the motions of the amino acid residues between (A)PD-L1 and (B)PD-L1; therefore, the deepest blue areas are on the diagonal of the figure [49,50]. The large red areas in the lower right corner of the graph indicated that the motion of the two chains has a strong negative movement. As for the blue areas, these atoms in the blue region are on the same chain, so a vast amount of the $C_{\alpha }$ atoms have the same direction in motion. In Figure 10, the red patch of the curcumin and 6-gingerol system occupies a larger area than those of the capsaicin and zucapsaicin system, which shows that the movements of (A)PD-L1 and (B)PD-L1 in the curcumin and 6-gingerol system are more flexible. These results are consistent with the capsaicin and zucapsaicin systems having lower binding free energy than the curcumin and 6-gingerol systems. In addition, the motion of the binding pocket is similar in the capsaicin and zucapsaicin system that is linked to the similarity of the key residues of capsaicin and zucapsaicin in the binding pocket. In addition, the residues of the binding site (C-sheet, F-sheet and G-sheet) between (A)PD-L1 and (B)PD-L1 have a large anti-correlated motion, indicating that these small molecules could interact with two chains of the PD-L1 dimer.

In short, the cross-correlation matrix results suggested that the food-derived molecules have a strong binding ability at the binding pocket in 200 ns MD simulation, and the complex systems exhibited more stable dynamic behaviors than the dimer system, owing to the binding of the food-derived compounds or their derivatives to the PD-L1 dimer.

### 2.8. Free Energy Landscape (FEL)

To further investigate the effect of these compounds on the conformational space of the PD-L1 dimer, we constructed the FEL of the small molecule system and the PD-L1 dimer system. The percentage of top 20 principal components (PCs) is shown in Appendix A.

It can be seen that from the perspective of overall movement, the intensity of protein movement of the small molecule systems in Figure 11 is smaller than that of the dimer system in Appendix A, in which the conformations of small molecule systems are more concentrated than the conformation of the dimer system. In addition, the small molecule system can induce very stable conformational regions in the 200 ns kinetic simulation, whereas the dimer system has larger motion space and is difficult to stabilize in the lower energy conformation in solvent. This indicated that there are more metastable states from conformation to conformation, and it is difficult for the dimer to exist in a stable state in the solvent [30]. It again confirms that these natural small molecules acting on the channel-like hydrophobic pocket of the PD-L1 dimer can stabilize the PD-L1 dimer and block the PD-1/PD-L1 pathway. Food-derived molecules and their derivatives can induce the dimerization of PD-L1, stabilize the channel structure by the hydrophobic groups (4′-hydroxy-3′-methoxyphenyl group et), and prevent the “beta folding surface” of PD-L1 to be exposed in the solvent and bind with PD-1, resulting in reducing the immune escape of cancer cells.

## 3. Materials and Methods

### 3.1. Molecular Docking and Initial Structure Construction

Molecular docking is a powerful tool in structural molecular biology and computer-assisted drug design to explore the predominant binding modes and predict the affinity of combination between ligands and proteins [51]. Although the outcome to a large extent has depended on protein–ligand interaction and molecular configuration [45], it could provide a ligand–protein complex structure for subsequent simulation research. The crystal structures of the PD-L1 dimer (PDBID: 5N2F) and ligands were obtained from the RCSB and ZINC library [8,52], respectively. Firstly, calculations of the small molecular models of capsaicin (ZINC1530575), zucapsaicin (ZINC4468952) and 6-gingerol (ZINC15331846) were carried out with the Gaussian09 software program [53] using the B3LYP functional with the London dispersion correction scheme of Grimme (denoted as D3) and def-TZVP basis sets to optimize the molecular structure [54,55,56]. In addition, the SPDBV program was used to fill the missing atoms of the PD-L1 dimer and remove the original ligand (BMS-200) [57]. Because the binding site was reported by Guo et al. [30], AutoDock Vina [58] was utilized to dock small molecules to receptors, where the box of 40 × 40 × 40 Å with default parameters covered the whole surface of the protein. Concurrently, the box was established with a 1.0 Å grid space centered at the binding pocket. In each docking experiment, the top complex structures were selected based on the binding affinity calculated by the AutoDock Vina scoring function, which analyzed the protein–ligand interaction and prepared for molecular dynamics simulations.

### 3.2. Molecular Dynamics Simulation

According to the docking result, the conformation of the complexes with the highest affinity was used for further MD simulation by the GROMACS 2016.4 package [59]. The charge part of small molecules was fitted the RESP, which was widely applied in the molecular simulations [31]. Then, small molecules and proteins were parameterized by the general AMBER force field (GAFF) and AMBER ff99SB [60,61], respectively, in which the small molecular top file was created by Mulitiwfn and the Sobtop program [62,63]. A cubic box with a side length of 10 Å was used to contain the complex structure, while TIP3P water [64] was added to fill the boxes so that the total number of atoms in each system was roughly the same, and sodium ions were added to make the system electrically neutral. Primarily, in order to eliminate undesirable interactions and atomic collisions, the steepest decent (SD) and conjugated gradient (CG) methods were implemented in the energy minimization step. Afterward, the temperature of these systems was raised from 0 to 300 K for 1 ns in the NVT ensemble, and a Berendsen thermostat was used to control the temperature of the protein–ligand group and solvent–ions group. A Parrinello–Rahman barostat was used to maintain 1 ns of equilibrium in the NPT ensemble at 1 atm pressure [12]. Finally, MD simulations were conducted for 200 ns, in which hydrogen bonds were constrained using the LINCS algorithm, and the short-range nonbonded interactions were computed with a cutoff of 10 Å. The Van der Waals interaction and electrostatic interaction were calculated by the cutoff and Particle Mesh Ewald (PME) [65] algorithms, respectively. Then, a V-rescale thermostat was used to maintain the temperature at 300 K, and the pressure was controlled by a Parrinello–Rahman barostat at 1 atm. Meanwhile, to check the stability of the systems and determine the statistical significance of the results, the simulations were performed thrice with the same parameters, and the trajectories were recorded every 10 picoseconds for subsequent analysis.

### 3.3. Binding Free Energy Calculation

Following the MD simulations, the binding free energies (∆G) were calculated using the gmxMMPBSA software package by the MM-PBSA method [32,33]; furthermore, a total of 500 snapshots from the final 50 ns stable MD trajectory were captured from all systems, which was utilized to obtain the binding free energy. Briefly, the MM-PBSA method can be summarized as follows:(1)$$\Delta G_{bind}=G_{complex}-G_{receptor}-G_{ligand}=\Delta H-T\Delta S$$
(2)$$G=E_{MM}+G_{sol}-TS$$
(3)$$\Delta H=\Delta E_{MM}+\Delta G_{sol}$$
where
(4)$$\Delta E_{MM}=\Delta E_{binded}+\Delta E_{nonbinded}=\;\left(\Delta E_{bind}+\Delta E_{angle}+\Delta E_{dihedral}\right)+\left(\Delta E_{ele}+\Delta E_{vdw}\right)$$
and
(5)$$\Delta G_{sol}=\Delta G_{polar}+\Delta G_{non\text{-}polar}=\Delta G_{PB/GB}+\Delta G_{non\text{-}polar}$$
where
(6)$$\Delta G_{non\text{-}polar}=NP_{TENSION}\times \Delta SASA+NP_{OFFSET}$$

In the above equations, the binding free energy ($\Delta G_{bind}$) is made of $G_{complex}$, $G_{recepter}$ and $G_{ligand}$ in Equation (1), which also can be represented as the sum of enthalpy of binding (ΔH) and conformational entropy after ligand binding (−TΔS). The free energy (G) in Equation (2) can be decomposed into $E_{MM}$, $G_{sol}$ and TS so that ΔH can be identified as consisting of $\Delta E_{MM}$ and $\Delta G_{sol}$ in Equation (3), where $\Delta E_{MM}$ and $\Delta G_{sol}$ corresponds to the molecular mechanical energy changes in the gas phase and the solvent energy, respectively. $\Delta E_{MM}$ includes $\Delta E_{binded}$, also known as internal energy, and $\Delta E_{nonbinded}$, which is further divided into the van der Waals interaction energy ($\Delta E_{vdw}$) and the electrostatic interaction energy ($\Delta E_{ele}$) in Equation (4). On the other hand, the solvation free energy ($\Delta G_{sol}$) is further divided into a polar ($\Delta G_{polar}$) and a non-polar ($\Delta G_{non\text{-}polar}$) component in Equation (5), in which the Poisson–Boltzmann model was utilized to compute the polar solvation component, and a SASA-only model was devoted to estimate the non-polar term, which was usually assumed to be proportional to the solvent accessible surface area (ΔSASA) of the receptor in Equation (6) with a proportionality constant derived from experimental solvation energies of non-polar molecules.

The TΔS term in Equation (1) was calculated by Interaction Entropy (IE), which is a highly efficient and accurate method without extra computational cost [66,67].
(7)$$T\Delta S=-KTln\langle e^{\beta \Delta E_{pl}^{int}}\rangle \;$$
(8)$$\langle e^{\beta \Delta E_{int}}\rangle =\frac{1}{N}\sum_{i=1}^{N}e^{\beta \Delta E_{int}\left(t_{i}\right)}$$
where β is 1/kT with k being the Boltzmann constant and $\Delta E_{int}=E_{int}-\Delta E_{int}\Delta$ in Equation (7), the thermal average of $e^{\beta \Delta E_{int}}$ can be carried out by time-averaging along the MD trajectory in Equation (8). Then, the restricted condition of IE was $\sigma \left(Int.\;Energy\right)$ < 3.6 kcal; otherwise, the results of IE could impossibly converge [68]. Furthermore, in order to obtain the residue energy decomposition of the two chains of dimers, gmxMMPBSA was employed to generate the per-residue energy to enable analyzing the impact of the key residues.

### 3.4. Simulation Analysis

Trajectory analysis was performed using the auxiliary tools provided by GROMACS 2016.4 package [59], in which the rms and rmsf module were utilized to evaluate the degree of structural change and the level of drastic fluctuation. The gyrate module was employed to calculate the radius of gyration, which describes the tightness of the receptor protein in the solvent. The Mindist module was applied to compute the contact numbers and interactions of the capsaicin, zucapsaicin and 6-gingerol systems, respectively. The Visual Molecular Dynamics (VMD) 1.9.3 software [69] was used to count the occupancies of H bonds in the complex systems, in which the conditions of H-bonds were an acceptor–hydrogen–donor angle > 135° and an acceptor–hydrogen–atom distance of <3.5 Å. In addition, PCA was performed to characterize the major motion of the PD-L1 dimer and the correlative motion between the atoms derived from the MD trajectories in complex systems [12,70]. However, before PCA, the translation and rotation of protein in the trajectory were eliminated by the auxiliary tools of GROMACS so that the intermolecular motion was not masked by the overall motion of protein. Then, the eigenvectors and the corresponding eigenvalues were generated by the COVAR module and using Cartesian coordinates of $C_{\alpha }$ atoms in order to describe the correlation of atomic motion of the PD-L1 dimer in different systems and the impact of ligands on conformational distribution [36,37].

Furthermore, the first principal component and the second principal component (PC1 and PC2) were generated to establish FEL as reaction coordinates, in which free energy was calculated by Equation (9):(9)G=−kT×lnP

k is Boltzmann’s constant, T is the temperature of the simulation systems, and P is the probability density of the various conformations.

## 4. Conclusions

In this work, we investigated the binding mechanism of food-derived small molecules and their derivatives (capsaicin, zucapsaicin, curcumin and 6-gingerol) to the PD-L1 dimer using an integration of semiflexible molecular docking, molecular dynamics simulation and free energy calculation. All of these molecules could interact with the PD-L1 dimer stably based on their negative binding free energies. Meanwhile, the intervention of food-derived molecules can augment the rigidity of the dimer to exist in the solvent with a relatively stable state. Fortunately, more effective small molecules such as zucapsaicin and capsaicin were demonstrated with stronger binding affinity to the PD-L1 dimer as compared to our previous finding. The binding free energy is following the trend: zucapsaicin > capsaicin > 6-gingerol ≈ curcumin. Notably, four key residues make crucial contributions to ligand binding (Ile54, Tyr56, Met115 and Ala121), as identified by per-residue energy decomposition and contact numbers analysis. Based on the analysis of binding modes and interactions, three key components were identified: the C, F and G sheets of the PD-L1 dimer. Specifically, the non-polar interactions between the 4′-hydroxy-3′-methoxyphenyl group of such compounds and the key residues of the PD-L1 dimer play a dominant role in enhancing their stability and affinity. The FEL results further imply that these compounds can interact stably with the binding regions of the PD-L1 dimer. Overall, this work offers an opportunity to identify available food-derived compounds and provide insights for designing new small molecules with a beneficial structural skeleton. Such structural features help to understand the druggable hotspots at the dimer interface and yield insights for developing food-derived molecules that directly target PD-L1 dimerization, thereby providing a potential solution for cancer immunotherapy. Further experimental work to verify the activity of the food-derived small molecules in this work is in progress.

## Figures and Tables

**Figure 1 ijms-24-01413-f001:**
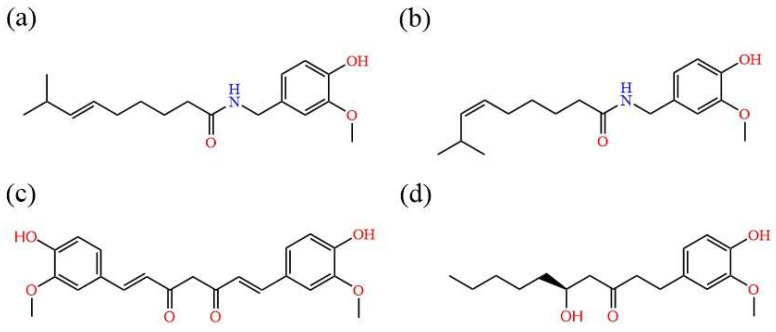
Molecular structural of (**a**) capsaicin, (**b**) zucapsaicin, (**c**) curcumin, (**d**) 6-gingerol.

**Figure 2 ijms-24-01413-f002:**
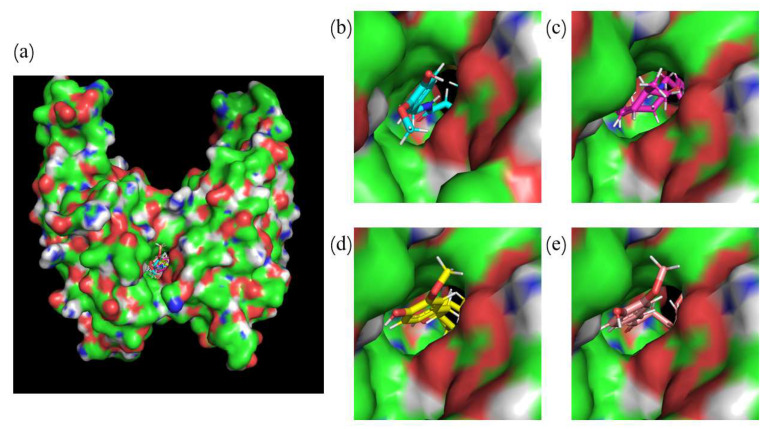
Initial structures of the systems used in MD simulations. (**a**) the binding site of small molecules targeting PD-L1 dimer, (**b**) initial structure of capsaicin, (**c**) initial structure of zucapsaicin, (**d**) initial structure of curcumin, and (**e**) initial structure of 6-gingerol.

**Figure 3 ijms-24-01413-f003:**
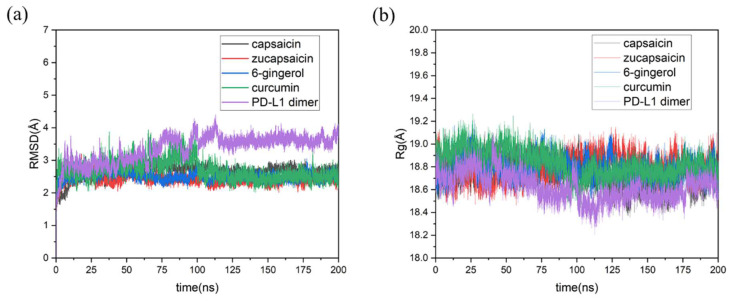
Estimation of MD simulation equilibration and analysis of the stability of dimer structure. Time evolutions of (**a**) the residues RMSD within 20 Å of binding site and (**b**) the radius of gyration ($R_{g}$) in protein.

**Figure 4 ijms-24-01413-f004:**
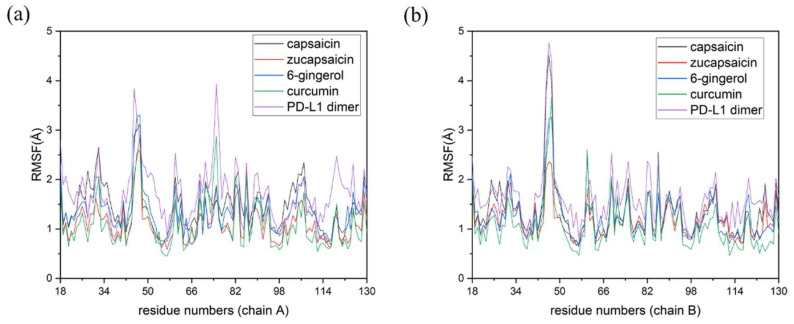
Root mean square fluctuation (RMSF) results of the MD simulations. (**a**) RMSF fluctuations of residues on (A)PD-L1. (**b**) RMSF fluctuations of residues on (B)PD-L1.

**Figure 5 ijms-24-01413-f005:**
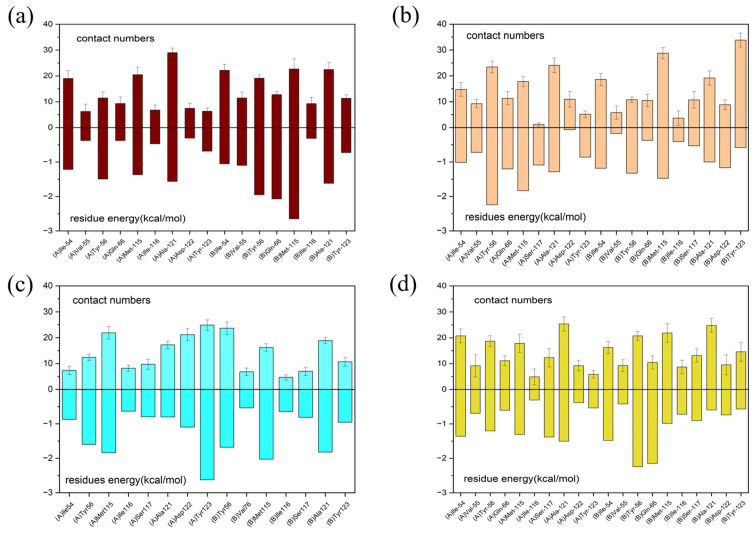
Residue energy decomposition of key residues and contact numbers between the small molecules and the PD-L1 dimer belonging to (**a**) capsaicin, (**b**) zucapsaicin, (**c**) curcumin and (**d**) 6-gingerol systems (kcal/mol).

**Figure 6 ijms-24-01413-f006:**
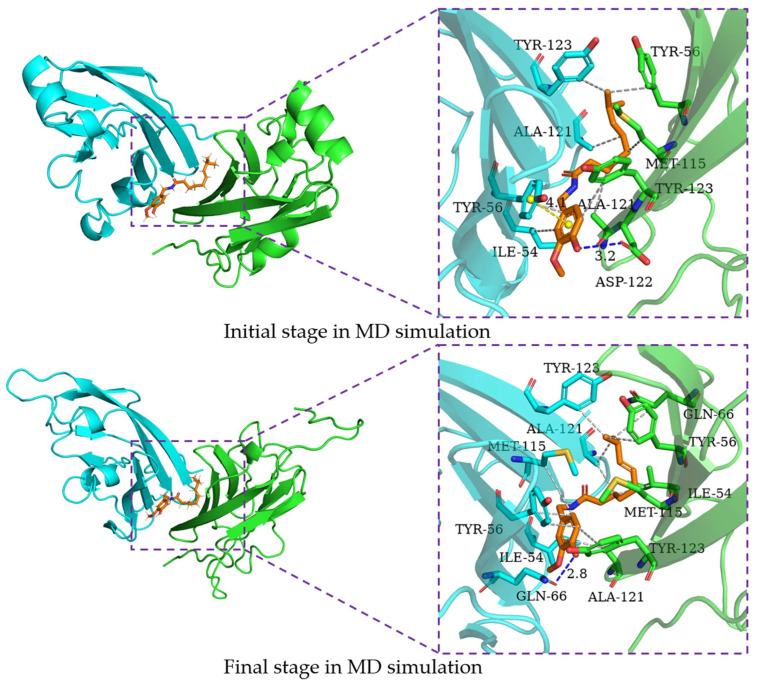
The interaction of capsaicin and residues of binding site at 0 ns and 200 ns, the key residues on (A)PD-L1 and (B)PD-L1 are shown as green and cyan sticks, respectively, while the ligands are shown as beige sticks. Hydrophobic interactions, H-bonds and Π-stacking are shown as gray, blue and yellow dashes, respectively.

**Figure 7 ijms-24-01413-f007:**
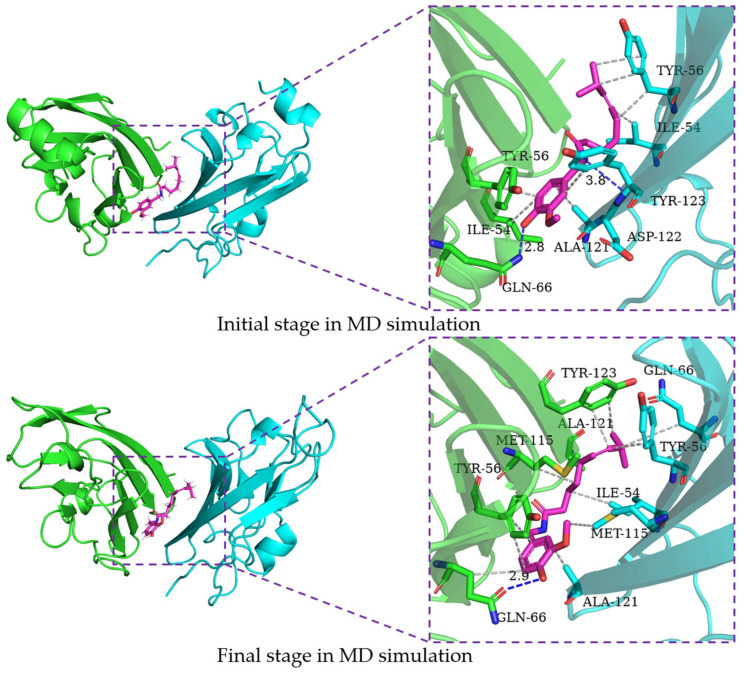
The interaction of zucapsaicin and residues of the binding site at 0 ns and 200 ns, the key residues on (A)PD-L1 and (B)PD-L1 are shown as green and cyan sticks, respectively, while the ligands are shown as pink sticks. Hydrophobic interactions, H-bonds and Π-stacking are shown as gray, blue and yellow dashes, respectively.

**Figure 8 ijms-24-01413-f008:**
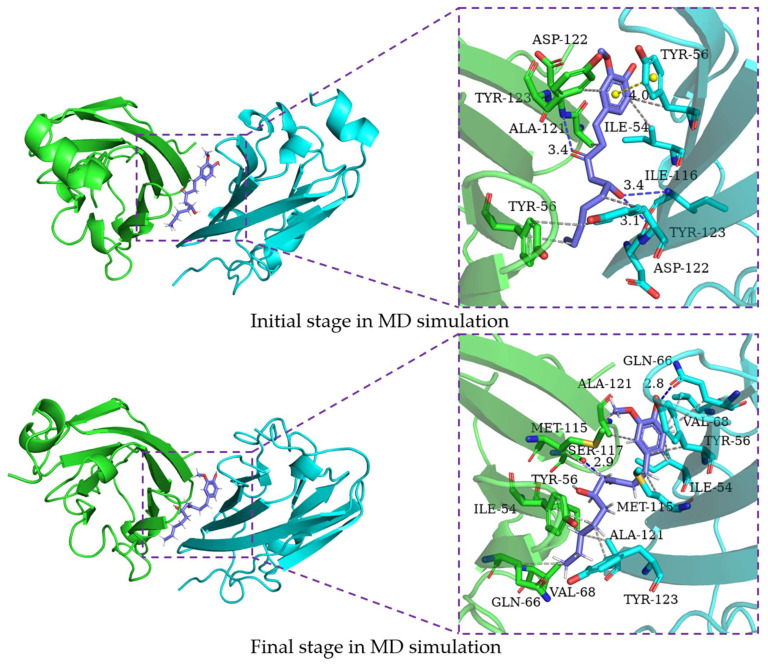
The interaction of 6-gingerol and residues of the binding site at 0 ns and 200 ns; the key residues on (A)PD-L1 and (B)PD-L1 are shown as green and cyan sticks, respectively, while the ligands are shown as purple sticks. Hydrophobic interactions, H-bonds and Π-stacking are shown as gray, blue and yellow dashes, respectively.

**Figure 9 ijms-24-01413-f009:**
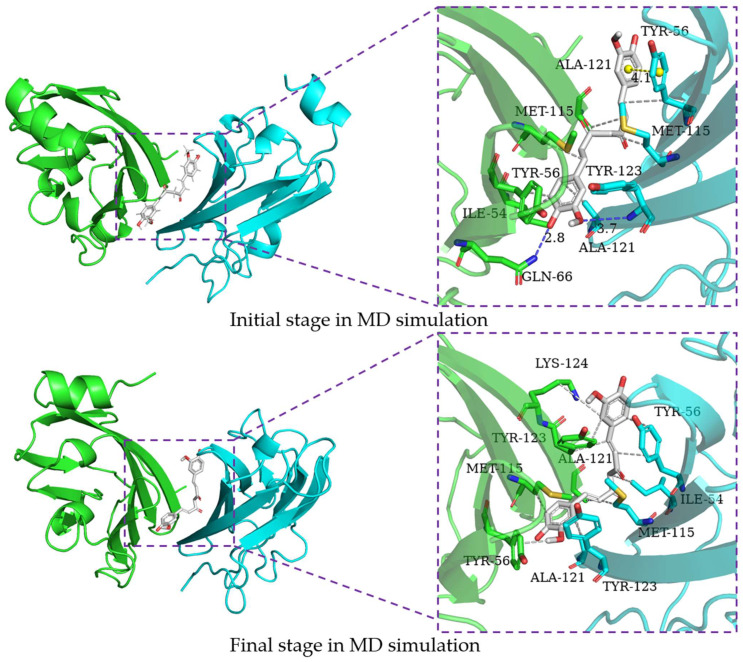
The interaction of curcumin and residues of binding site at 0 ns and 200 ns, the key residues on (A)PD-L1 and (B)PD-L1 are shown as green and cyan sticks, respectively, while the ligands are shown as white sticks. Hydrophobic interactions, H-bonds and Π-stacking are shown as gray, blue and yellow dashes, respectively.

**Figure 10 ijms-24-01413-f010:**
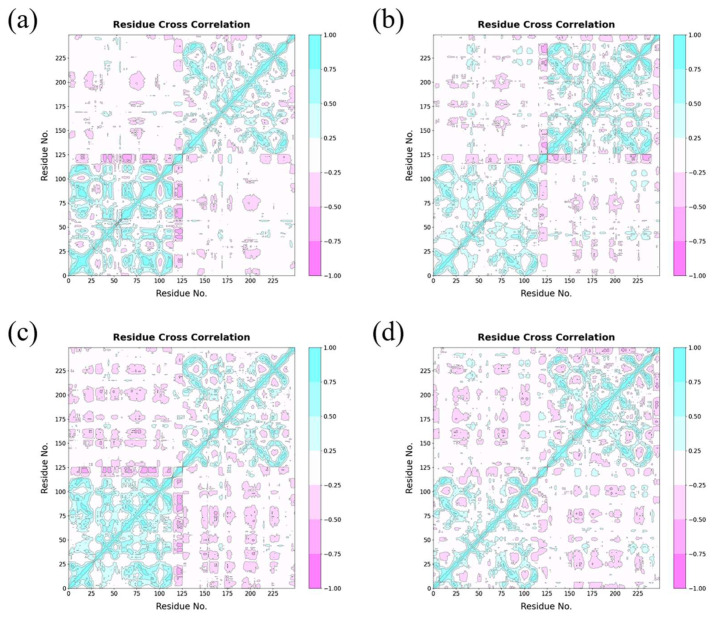
Cross-correlation matrixes of fluctuations in the x-, y- and z-coordinates for $C_{\alpha }$ atoms belonging to the PD-L1 dimer in the (**a**) capsaicin, (**b**) zucapsaicin, (**c**) curcumin and (**d**) 6-gingerol systems.

**Figure 11 ijms-24-01413-f011:**
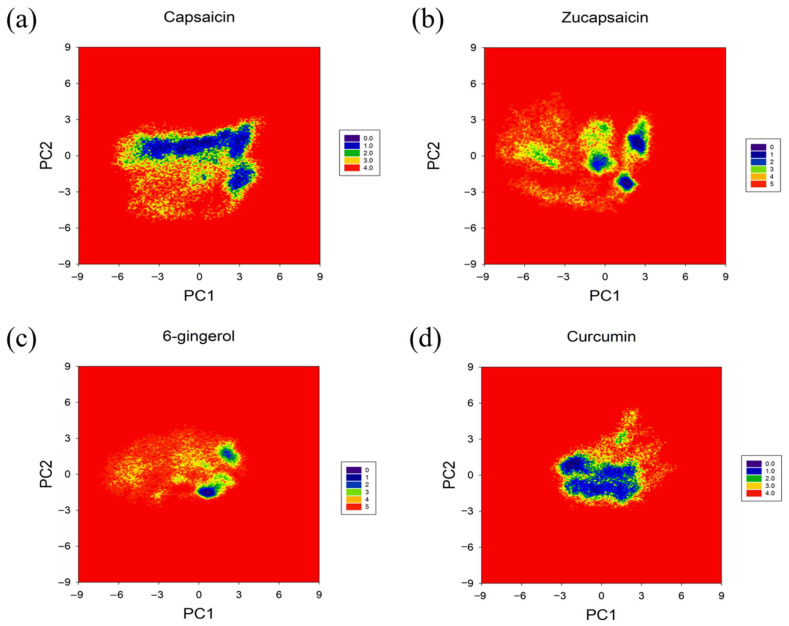
Free energy landscapes (KT) of the (**a**) capsaicin, (**b**) zucapsaicin, (**c**) curcumin and (**d**) 6-gingerol systems. PC1 and PC2 represent principal components 1 and 2, respectively.

**Table 1 ijms-24-01413-t001:** Binding free energies in the capsaicin, zucapsaicin, curcumin and 6-gingerol system (kcal/mol).

Component (kcal/mol)	Capsaicin	Zucapsaicin	Curcumin	6-Gingerol
$\Delta E_{vdmwaals}$	−56.86 ± 3.14	−55.41 ± 2.56	−55.81 ± 2.40	−52.45 ± 3.02
$\Delta E_{ele}$	−4.76 ± 1.44	−3.06 ± 2.19	−4.14 ± 1.50	−7.06 ± 2.79
$\Delta E_{polar}$	15.88 ± 1.10	15.25 ± 1.60	19.02 ± 1.57	15.27 ± 1.72
$\Delta E_{nonpolar}$	−4.29 ± 0.13	−4.40 ± 0.12	−4.74 ± 0.09	−4.22 ± 0.03
$\Delta G_{gas}$	−61.62 ± 3.28	−58.46 ± 2.89	−59.94 ± 2.93	−58.12 ± 3.20
$\Delta G_{solvent}$	11.59 ± 1.10	10.80 ± 1.59	14.27 ± 1.56	11.05 ± 1.71
TΔS	7.50 ± 0.87	3.91 ± 0.08	6.48 ± 0.05	7.87 ± 0.05
$\sigma \left(Int.\;Energy\right)$	3.28	2.93	3.20	2.89
$\Delta G_{bind}$	−42.53 ± 3.30	−43.75 ± 2.58	−39.19 ± 2.55	−39.20 ± 2.66

**Table 2 ijms-24-01413-t002:** The components of binding free energies of the PD-L1 monomer/capsaicin, PD-L1 monomer/zucapsaicin, PD-L1 monomer/curcumin and PD-L1 monomer/6-gingerol systems (kcal/mol).

Component (kcal/mol)	PD-L1 Monomer/Capsaicin	PD-L1 Monomer/Zucapsaicin	PD-L1 Monomer/Curcumin	PD-L1 Monomer/6-Gingerol
$\Delta E_{vdmwaals}$	−16.06 ± 5.43	−24.61 ± 5.02	−29.37 ± 3.89	−20.89 ± 4.62
$\Delta E_{ele}$	−1.66 ± 0.46	−2.80 ± 1.94	−4.21 ± 2.80	−6.03 ± 4.65
$\Delta E_{polar}$	5.23 ± 3.55	7.30 ± 2.55	10.15 ± 3.02	8.91 ± 2.24
$\Delta E_{nonpolar}$	−2.06 ± 0.99	−2.67 ± 0.40	−2.86 ± 0.36	−2.44 ± 0.40
$\Delta G_{gas}$	−17.73 ± 6.36	−27.41 ± 5.50	−33.57 ± 5.52	−26.93 ± 6.32
$\Delta G_{solvent}$	3.17 ± 1.89	4.63 ± 2.44	7.29 ± 2.80	6.47 ± 0.62
$\Delta G_{bind}$	−14.56 ± 5.52	−22.79 ± 4.53	−26.28 ± 3.47	−20.46 ± 4.30

**Table 3 ijms-24-01413-t003:** Capsaicin system per-residue energy decomposition.

Residue Number	Van Der Waals (kcal/mol)	Ele (kcal/mol)	PB (kcal/mol)	Total (kcal/mol)	Contact Number
(A)Ile-54	−1.333	−0.127	0.246	−1.215	18.98 ± 3.12
(A)Val-55	−0.523	0.062	0.084	−0.376	6.22 ± 2.87
(A)Tyr-56	−1.648	−0.059	0.214	−1.495	11.43 ± 2.31
(A)Gln-66	−0.713	0.006	0.084	−0.376	9.31 ± 2.53
(A)Met-115	−1.945	−0.185	0.761	−1.370	20.47 ± 2.92
(A)Ile-116	−0.674	0.060	0.142	−0.471	6.79 ± 1.92
(A)Ala-121	−1.973	−0.108	0.517	−1.563	28.98 ± 1.79
(A)Asp-122	−0.786	0.247	0.231	−0.307	7.47 ± 1.92
(A)Tyr-123	−0.665	−0.102	0.083	−0.684	6.31 ± 1.30
(B)Ile-54	−1.992	0.873	0.065	−1.054	22.15 ± 2.35
(B)Val-55	−1.108	−0.419	0.431	−1.096	11.43 ± 2.31
(B)Tyr-56	−2.164	0.083	0.130	−1.951	19.12 ± 1.28
(B)Gln-66	−0.965	−2.317	1.211	−2.072	12.77 ± 1.21
(B)Met-115	−1.996	−1.537	0.882	−2.651	22.68 ± 3.98
(B)Ile-116	−1.033	0.551	0.167	−0.315	9.29 ± 2.44
(B)Ala-121	−1.131	−0.620	0.133	−1.618	22.48 ± 2.62
(B)Tyr-123	−0.944	0.025	0.191	−0.728	11.31 ± 1.42

**Table 4 ijms-24-01413-t004:** Zucapsaicin per-residue energy decomposition.

Residue Number	Van Der Waals (kcal/mol)	Ele (kcal/mol)	PB (kcal/mol)	Total (kcal/mol)	Contact Number
(A)ILE-54	−0.826	0.053	−0.014	−0.787	14.75 ± 2.62
(A)Tyr-56	−2.841	−0.254	0.391	−2.703	23.44 ± 2.23
(A)Arg-113	−1.945	−0.185	0.761	−1.370	16.64 ± 2.17
(A)Met-115	−1.608	−0.443	0.486	−1.565	17.84 ± 1.94
(A)Ser-117	−0.123	−0.021	0.059	−0.084	1.72 ± 0.71
(A)Tyr-123	−0.793	−0.361	0.486	−0.668	5.14 ± 1.27
(B)Tyr-56	−1.563	0.119	0.489	−0.954	10.78 ± 1.03
(B)Trp-57	−0.417	−0.101	0.007	−0.512	3.13 ± 0.94
(B)Arg-113	−2.202	−0.015	−0.130	−2.318	16.36 ± 0.95
(B)Cys-114	−0.928	−0.804	0.390	−1.342	7.29 ± 0.83
(B)Met-115	−2.625	−0.071	0.249	−2.447	28.73 ± 2.18
(B)Ala-121	−1.084	−0.100	1.008	−0.175	19.17 ± 2.77
(B)Tyr-123	−3.199	−0.954	0.722	−3.431	33.83 ± 2.72

**Table 5 ijms-24-01413-t005:** Curcumin per-residue energy decomposition.

Residue Number	Van Der Waals (kcal/mol)	Ele (kcal/mol)	PB (kcal/mol)	Total (kcal/mol)	Contact Number
(A)Ile-54	−0.685	−0.298	0.112	−0.871	7.33 ± 1.59
(A)Tyr-56	−1.459	−0.379	0.237	−1.601	12.41 ± 1.23
(A)Met-115	−2.315	−0.259	0.736	−1.839	21.89 ± 2.40
(A)Ile-116	−0.841	−0.111	0.317	−0.634	8.18 ± 1.14
(A)Ser-117	−0.775	−0.143	0.124	−0.794	9.70 ± 1.97
(A)Ala-121	−1.287	0.214	0.269	−0.803	17.24 ± 1.43
(A)Asp-122	−1.911	−2.262	3.077	−1.095	21.18 ± 2.29
(A)Tyr-123	−2.998	−1.095	1.469	−2.624	24.88 ± 2.03
(B)Tyr-56	−2.467	0.113	0.671	−1.682	23.65 ± 2.38
(B)Val-76	−0.489	−0.069	0.019	−0.539	6.80 ± 1.47
(B)Met-115	−1.505	−0.896	0.376	−2.025	16.22 ± 1.47
(B)Ile-116	−0.717	−0.333	0.405	−0.645	4.66 ± 1.01
(B)Ser-117	−0.692	−0.142	0.022	−0.813	6.98 ± 1.63
(B)Ala-121	−1.432	−0.841	0.451	−1.823	18.90 ± 1.24
(B)Tyr-123	−1.131	0.136	0.038	−0.955	10.68 ± 1.71

**Table 6 ijms-24-01413-t006:** 6-Gingerol per-residue energy decomposition.

Residue Number	Van Der Waals (kcal/mol)	Ele (kcal/mol)	PB (kcal/mol)	Total (kcal/mol)	Contact Number
(A)ILE-54	−1.425	−0.111	0.173	−1.362	20.73 ± 2.69
(A)Val-55	−0.734	−0.249	0.289	−0.693	9.15 ± 4.31
(A)Tyr-56	−1.405	0.046	0.154	−1.204	18.67 ± 2.14
(A)Gln-66	−0.944	−0.094	0.434	−0.604	11.14 ± 1.98
(A)Met-115	−1.568	−0.245	0.508	−1.305	17.84 ± 3.47
(A)Ile-116	−0.585	0.126	0.150	−0.308	4.88 ± 3.06
(A)Ser-117	−0.754	−1.228	0.601	−1.382	12.32 ± 3.58
(A)Ala-121	−1.694	−0.103	0.294	−1.502	25.34 ± 2.74
(A)Asp-122	−0.937	−0.010	0.563	−0.384	9.21 ± 2.02
(A)Tyr-123	−0.575	0.031	0.005	−0.537	5.84 ± 1.44
(B)Ile-54	−1.483	−0.271	0.276	−1.478	16.21 ± 2.41
(B)Val-55	−0.932	0.268	0.242	−0.421	9.22 ± 2.31
(B)Tyr-56	−2.385	−0.126	0.272	−2.240	20.74 ± 1.72
(B)Gln-66	−0.647	−2.810	1.318	−2.148	10.44 ± 2.52
(B)Met-115	−1.745	−0.003	0.760	−0.988	21.84 ± 3.64
(B)Ile-116	−0.846	−0.186	0.310	−0.721	8.68 ± 2.65
(B)Ser-117	−0.831	−0.373	0.300	−0.904	13.11 ± 2.55
(B)Ala-121	−1.212	0.170	0.445	−0.596	24.75 ± 2.70
(B)Asp-122	−0.644	−0.746	0.651	−0.738	9.51 ± 3.87
(B)Tyr-123	−0.826	0.072	0.182	−0.571	14.55 ± 3.64

**Table 7 ijms-24-01413-t007:** The H-bonds occupancy of capsaicin, zucapsaicin, curcumin and 6-gingerol (-side is the sidechain of residue, -main is the mainchain of residue).

H-Bonds Donor	H-Bonds Acceptor	H-Bonds Occupancy	Average Numbers of H-Bonds
Capsaicin-O21	(B)Gln-66-side-OE1	91.40%	19
Capsaicin-O21	(A)Ala-121-main-O	41.12%	
(A)Ile-116-main-N	Capsaicin-O22	24.35%	
(B)Gln-66-side-NE2	Capsaicin-O20	11.58%	
(A)Ser-117-main-N	Capsaicin-O22	10.18%	
Zucapsaicin-O3	(A)Gln-66-side-NE2	29.34%	14
Zucapsaicin-N4	(B)Met-115-side-SD	23.15%	
Zucapsaicin-O3	(B)Ala-121-main-O	36.73%	
Zucapsaicin-O3	(A)Asp-73-side-OD1	19.56%	
Zucapsaicin-O3	(A)Gln-66-side-OE1	36.53%	
(B)Ile-116-main-N	Zucapsaicin-O1	17.37%	
(B)Ser-117-main-N	Zucapsaicin-O1	22.55%	
(B)Ser-117-side-OG	Zucapsaicin-O1	11.38%	
6-gingerol-O19	(B)Gln-66-side-OE1	85.03%	25
(B)Ser-117-side-OG	6-gingerol-O20	56.11%	
(B)Gln-66-side-NE2	6-gingerol-O19	13.97%	
6-gingerol-O19	(B)Ala-121-main-O	37.33%	
6-gingerol-O21	(B)Met-115-main-O	21.16%	
6-gingerol-O21	(B)Ile-54-main-O	16.97%	
(B)Ile-116-main-N	6-gingerol-O21	14.57%	
Curcumin-O27	Tyr-56-side-OH	13.37%	18
(A)Tyr-123-main-N	Curcumin-side-O23	55.69%	
Curcumin-O22	(A)Ala121-main-O	20.16%	
(A)Asp-122-main-N	Curcumin-side-O23	35.33%	
(A)Lys-124-side-NZ	Curcumin-O27	20.56%	
(B)Ile-116-main-N	Curcumin-O24	43.71%	

## Data Availability

The data presented in this study are available within the article, figures, tables and Appendix A.

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
