# Peer review of "Is the Triggering of PD-L1 Dimerization a Potential Mechanism for Food-Derived Small Molecules in Cancer Immunotherapy? A Study by Molecular Dynamics"

_ijms, 2023, doi:10.3390/ijms24021413_

Round 1

Reviewer 1 Report

The introduction section is not clear. It is very broad and not at all comprehensively focused to relevance of the manuscript. It lists a lot of other people's research work, but it does not impress on readers the importance of this work, what is not solved in this field and why is it important. Therefore, as a reader, it is difficult to draw the conclusion from them as to why this study has been carried out. The authors need to discuss the previous work instead of only mentioning that author `A' did this and author `B' did this. In conclusion, there is not any insight into the physical description of the problem studied beyond the determination of a number of parameters by means of computing software

Reviewer 2 Report

Wu et al. used molecular modeling and molecular simulations to investigate the role of four food-derived small molecules in PD-L1 inhibition. 

Although the authors provided convincing evidence for the capability of the four molecules to bind the PD-L1 dimer, I do not believe it is sufficient to support the authors' claim that the drug molecules can stabilize the PD-L1 dimerization. In this regard, I have the following questions/comments:

(1) The authors showed the RMSD of the protein dimer is the largest without a ligand. Based on this and other analysis, the authors claim that dimer becomes less stable when drug molecules are absent. However, the simulations were initialed from the holo structure which is biased toward a conformational state where the protein is more stable with a bound molecule. The authors should perform the simulations from the equilibrated apo state as well and perform the same analysis. Or equilvalently, the authors should elongate

theirs simulations and preform the analysis on the equilibrated part only.

(2) The key conclusion of the work is that the drug molecules can stabilize the PD-L1 dimer, but the authors can only show that the drug molecules can stably bind to the PD-L1 dimer. These two statements are fundamentally different and to prove the former statement, one needs to show that the free energy difference between dimer and two-monomers becomes more negative when the drug molecule is present. So, the drug binding free energy to the monomer must also be computed if one would like to compute this ΔΔG using a thermodynamic cycle.

Reviewer 3 Report

Thank you for the opportunity to review this manuscript, dealing with interesting findings entitled “Is the triggering of PD-L1 dimerization a potential mechanism for food-derived small molecules in cancer immunotherapy? A study by molecular dynamics”. They proposed that food-derived small molecules and their derivatives may serve as potential

candidates in cancer immunotherapy. Their bioinformatics observation/study aimed to understand how these compounds could inhibit PD-1/PD-L1 interactions by directly targeting PD-L1 dimerization. All findings are interesting, and the article includes a balanced and critical view of the findings. However, there are a few reports that not only have similar findings with some similar small molecules’ use but also approaches (Int. J. Mol. Sci. 2021, 22, 10924). In addition, these reports' citation is missing in this article. Though there's a difference in some small molecules. I am wondering if the manuscript passes the journal's plagiarism criteria since the way of writing including sentences are similar in several places. I would recommend checking this manuscript for plagiarism thoroughly to confirm whether it comes under the journal criteria regarding copyright. Once it clears, authors need to add citations of all published reports with similar approaches/findings. Furthermore, the authors need to discuss previous findings briefly in the discussion section.

Round 2

Reviewer 1 Report

Reasonable and necessary explanations are provided for the comments of referees. Hence now I do support publication of this manuscript for publication.

Reviewer 2 Report

The authors have addressed my questions in the response and revised manuscript.

Reviewer 3 Report

The revised manuscript and the author's response are satisfactory. The manuscript entitled "Is the triggering of PD-L1 dimerization a potential mechanism for food-derived small molecules in cancer immunotherapy? A study by molecular dynamics" can be published in the IJMS in its current stage.

Thank you.